# Generic Medical Image Segmentation Enhancement by Adapting Segment Anything Model

## Abstract

Accurate medical image segmentation is crucial for clinical applications but remains challenging due to ambiguous boundaries, multi-scale anatomies, and the high cost of expert annotations. While deep learning models often produce coarse initial masks, enhancing them into clinically reliable outputs is a critical yet under-explored problem. We propose SAMedEnhancer, a generic medical image segmentation enhancement framework that enhances coarse masks from any segmentation model using a strategically adapted Segment Anything Model (SAM). Our key innovation is a morphology-aware prompt generation strategy. It first analyzes initial masks via connected-component and shape analysis to identify reliable anatomical regions. Then, a hierarchical prompting mechanism is devised: positive points are sampled from high-confidence interiors, while negative points are selected from informative nearby backgrounds within dilated regions; these are supplemented by bounding boxes enclosing the refined targets. This coarse-to-fine prompting robustly guides SAM to recover accurate boundaries, resisting error propagation from imperfect inputs. We extensively validate SAMedEnhancer on a comprehensive benchmark for medical image segmentation enhancement, encompassing several datasets across various imaging modalities and both fully- and semi-supervised settings. Results demonstrate that our method consistently improves segmentation quality from state-of-the-art segmenters, reduces annotation dependency, and serves as a versatile accelerator for medical image segmentation.

## 1 Introduction

Accurate segmentation of anatomical structures and pathologies in medical imaging is fundamental to a wide range of clinical applications, including diagnostic support, treatment planning, and quantitative disease monitoring. Recent advances in deep learning have enabled automated and efficient segmentation, significantly advancing the field (Tang et al., 2019; Ronneberger et al., 2015; Zhou et al., 2018; Chen et al., 2024). However, the segmentation performance remains constrained by the limited availability of large-scale annotated datasets and the high cost associated with obtaining fine-grained labels from domain experts (Chen et al., 2024; Qi et al., 2023). While semi-supervised (Chen et al., 2023; Wang & Li, 2023), weakly supervised Kuang et al. (2023); Girum et al. (2020), or un-supervised learning approaches (Liu et al., 2023) have emerged to alleviate annotation scarcity, the quality of the resulting pseudo-labels remains a critical bottleneck (Chen et al., 2023; Wang & Li, 2023). These labels often suffer from boundary errors, fragmentation, and anatomical inaccuracies that ultimately constrain segmentation performance and clinical utility.

Enhancement-based approaches have emerged as a promising direction for improving segmentation quality (Patil et al., 2017; Pal et al., 2019; Lin et al.). Such approaches seek to fuse coarse segmentation priors to guide more precise delineation of complex anatomies (Qi et al., 2023), refine pseudo-labels to enhance the performance of semi- or unsupervised segmenters (Chen et al., 2021), or directly post-process network prediction probabilities (Larrazabal et al., 2020; Li et al., 2024). Despite their potential, existing enhancement techniques exhibit notable shortcomings. Classical morphological operations (*e.g.*, opening and closing, level-set) can eliminate small artifacts and smooth contours but operate solely on low-level pixel aggregates without semantic or anatomical awareness, often resulting in over-simplified or even anatomically erroneous outputs (Patil et al., 2017; Pal et al.,

2019). Model-dependent refiners, typically trained end-to-end with specific segmentation models, may achieve notable gains yet lack generalizability across architectures and tasks (Li et al., 2023a). On the other hand, model-agnostic methods, such as conditional random fields (CRFs) (Kamnitsas et al., 2017; Atif et al., 2019), shape-based filters, and heuristic rule sets, avoid retraining but often rely on hand-crafted features and carefully tuned hyperparameters optimized for specific anatomies or imaging modalities (Larrazabal et al., 2020; Kim & Kang, 2021). As a result, these methods often struggle to generalize across diverse segmentation tasks and may introduce new artifacts when applied to out-of-domain scenarios.

Recently, the emergence of interactive segmentation models, particularly the Segment Anything Model (SAM) (Kirillov et al., 2023), has introduced a new paradigm for generating high-fidelity object masks through user-provided prompts such as points and bounding boxes. SAM's strong visual representation and zero-shot generalization ability offer considerable potential to address refinement challenges in medical imaging (Cheng et al., 2023; Zhu et al., 2024; Ma et al., 2025; Wu et al., 2025). Specifically, enhancing coarse initial masks, rather than segmenting from scratch, could markedly reduce the dependency on large annotated datasets (Lin et al.). A natural idea is to leverage imperfect coarse masks to automatically generate prompts to guide SAM-based refinement. However, SAM's segmentation quality is highly sensitive to the precision of these prompts. Noisy or erroneous regions in coarse predictions, such as boundary deviations, false positives, or false negatives, can mislead prompt extraction, causing error amplification rather than correction. For instance, a bounding box tightly fitted to a coarse mask might include inaccurately segmented regions, leading SAM to reinforce existing mistakes. Similarly, point prompts sampled near ambiguous boundaries could steer the model towards incorrect edges. Therefore, the key challenge lies in how to reliably extract noise-robust anatomical prompts from imperfect coarse masks to effectively harness SAM's enhancement capability without propagating inherent inaccuracies.

In this paper, we introduce SAMedEnhancer, a generic enhancement framework that effectively adapts the Segment Anything Model (SAM) for high-precision medical image segmentation refinement. The core innovation lies in a morphology-aware prompting strategy that is highly robust to noise and inaccuracies commonly present in coarse segmentation masks. Our approach begins with a Morphological Split-Filter-Fuse step, which decomposes the coarse mask into connected components, filters out anatomically implausible fragments, and fuses semantically consistent regions. Building on these cleaned regions, we introduce a Hierarchical Prompt Excavation mechanism: first, positive and negative point prompts are intelligently sampled from high-confidence interior and background areas using distance transforms; then, adaptive bounding boxes are derived to tightly enclose the target structures while preserving contextual information. This coarse-to-fine prompting strategy effectively guides SAM to reconstruct accurate, topologically consistent boundaries, even when the input masks are significantly imperfect. To support comprehensive evaluation, we construct a large-scale medical mask enhancement benchmark comprising several public datasets across multiple imaging modalities and evaluate under both fully- and semi-supervised settings. Furthermore, we introduce a novel semi-supervised learning strategy that iteratively applies SAMedEnhancer to refine pseudo-labels, substantially reducing annotation dependency while improving segmentation performance. Our main contributions are summarized as follows:

- We propose SAMedEnhancer, the first framework to effectively leverage SAM for generic medical image segmentation enhancement by morphological analysis and hierarchical prompt excavation. It operates in a plug-and-play manner, refining coarse outputs from any segmentation model without retraining.

- We establish a large-scale evaluation benchmark for medical image segmentation enhancement, covering multiple datasets, modalities, and supervision settings, to facilitate future research in this underexplored area.

- We introduce a novel semi-supervised learning strategy that integrates iterative pseudo-label refinement with SAMedEnhancer, demonstrating significant performance gains and reduced reliance on annotated data.

## 2 RELATED WORK

### 2.1 ENHANCEMENT-BASED METHODS FOR MEDICAL IMAGE SEGMENTATION

Enhancement-based methods aim to enhance the low-quality outputs, either through post-processing raw predictions, correcting pseudo-labels, or integrating structural priors to guide precise delineation. Existing approaches can be broadly categorized into three classes: traditional low-level operations, model-specific learned enhancers, and model-agnostic heuristic methods.

Traditional approaches operate on low-level pixel or regional aggregates, leveraging geometric or statistical properties to improve segmentation smoothness and consistency, without incorporating semantic understanding of anatomical structures or clinical context. Common techniques include morphological operations (Patil et al., 2017; Pal et al., 2019), level-set (Li et al., 2010), and probabilistic graphical models like Markov Random Fields (Paragios et al., 2016) or Conditional Random Fields (Kamnitsas et al., 2017; Atif et al., 2019; He et al., 2004).

Model-dependent methods typically train end-to-end with specific segmentation models. These methods involve training a refinement network for integrating structural priors to guide precise segmentation or directly training a post-processing network. Li et al. (2023a) introduced a specialized network trained to identify and correct disrupted topology in initial segmentations by computing Euler characteristics for local image patches. While these learned refiners can achieve notable performance gains on their specific target data, their critical limitation is a lack of generalizability.

Model-agnostic refinement techniques enhance segmentation predictions independently of the base model's architecture but are often tailored to specific tasks. For instance, Post-DAE incorporates shape and topological priors to enhance outputs from any classifier, yet requires training a denoising autoencoder on segmentation masks (Larrazabal et al., 2020). Similarly, Li et al. (2024) trained a topology enhancement network on synthetic data encompassing diverse topological errors. While these approaches are model-agnostic, they still depend on training a specialized enhancement model using significant domain knowledge. An alternative, proposed by Kim & Kang (2021), is a recursive feedback mechanism that operates without training data or domain-specific knowledge. However, it lacks high-level semantic context, which hinders performance in complex scenarios.

### 2.2 SEGMENT ANYTHING MODEL

The Segment Anything Model (Kirillov et al., 2023) represents a milestone in promotable image segmentation, demonstrating remarkable zero-shot generalization across diverse domains. In medical imaging, early efforts primarily focused on fine-tuning SAM for specific tasks (Cheng et al., 2023; Deng et al., 2023; Zhu et al., 2024; Ma et al., 2025; Wu et al., 2025), or explored few/zero-shot settings (Ding et al., 2023; Li et al., 2023b; Wu & Xu, 2024; Butoi et al., 2023) to reduce annotation reliance, yet most still require manual interaction or task-specific samples. However, SAM's potential in medical segmentation enhancement, a core need for correcting coarse masks, and the efficacy of anatomy-tailored prompting strategies remain underexplored. Our work bridges this gap by introducing a robust prompting strategy that effectively harnesses SAM's capabilities for enhancing imperfect medical masks across diverse modalities and tasks.

## 3 METHOD

### 3.1 OVERVIEW

We propose **SAMedEnhancer**, a generic medical image segmentation refinement framework tailored for enhancing medical image segmentation by leveraging the medical-adapted Segment Anything Model. SAMedEnhancer is model-agnostic (*i.e.*, plug-and-play for any base segmentation model) and requires no additional training on target tasks, addressing the generalization limitations of existing enhancement methods.

The core of our approach is a novel morphology-aware prompting strategy that robustly converts noisy initial predictions into high-quality prompts to guide SAM, even when the input masks contain significant inaccuracies. As illustrated in Fig. 1, our method operates in three stages: (1) *Morphological Split-Filter-Fuse* extracts anatomically plausible and semantically coherent regions from

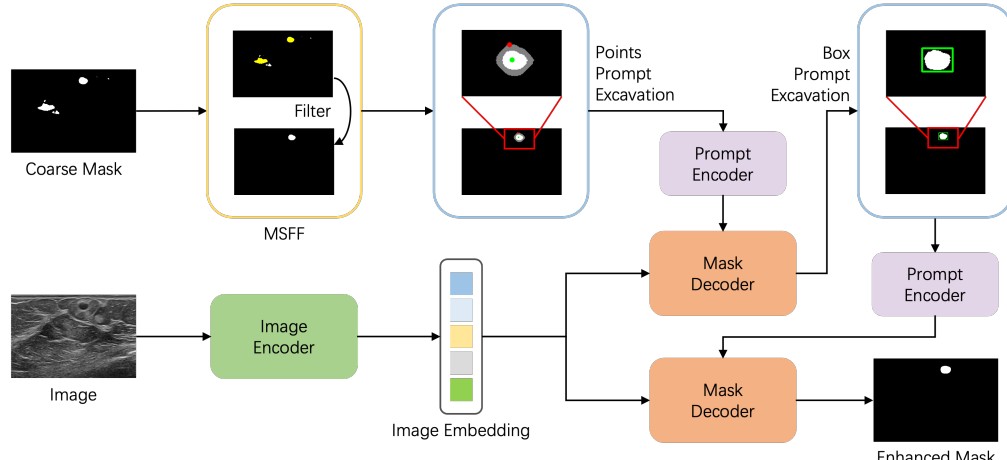

Figure 1: Overview of the SAMedEnhancer. SAMedEnhancer leverages SAM to refine medical coarse masks by automatically generating prompts from coarse masks, including a Morphological Split-Filter-Fuse (MSFF) module and Hierarchical Points-Box Prompt Excavation. The details of MSFF are illustrated in Algorithm 1.

coarse segmentation; (2) *Hierarchical Prompt Excavation* is proposed to extract robust and informative prompts hierarchically for enhancing targets from coarse-to-fine. The entire framework is model-agnostic and can be seamlessly integrated into various segmentation pipelines. The details will be illustrated in the following sections.

## 3.2 MORPHOLOGICAL SPLIT-FILTER-FUSE

Prompt quality is critical for SAM-based segmentation. However, coarse segmentation masks often contain spurious noise regions, fragmented structures, and regions corresponding to multiple objects. These artifacts can lead to unreliable or misleading prompts, ultimately degrading segmentation performance. To extract anatomically plausible and semantically coherent regions from such imperfect inputs, we propose a Morphological Split-Filter-Fuse (MSFF) module that incorporates morphological operations and domain-specific priors.

The MSFF process consists of three stages. 1) Split: For each target class, the predicted mask is decomposed into all connected components. These regions may include noise, fragmented structures, or correspond to multiple object instances. 2) Filter: The connected regions are filtered based on criteria such as size and morphological properties. For instance, for targets with regular shapes (*e.g.*, cells or specific organs), the Isoperimetric Quotient (IPQ), a metric that quantifies shape compactness, is employed to select anatomically plausible regions from the filtered components. 3) Fuse: To form semantically meaningful regions, similar components are iteratively merged based on the change in bounding box area and the mask area occupancy of the bounding box. Two components are merged only if the bounding box area change (before and after merging) is minimal and the mask area occupancy of the merged bounding box is sufficiently high (Lin et al.). Additionally, domain-specific priors can be incorporated into the merged regions to impose constraints. For example, in cell segmentation, multiple cells of the same type may be output from a single image; in contrast, for specific organ segmentation, only one target organ is output.

MSFF enables the formation of more reasonable single or multiple semantically meaningful regions, which serve as a reliable foundation for subsequent prompt generation. The details of the process are illustrated in Algorithm 1. The MSFF procedure is not fixed but flexible and can be adapted to the segmentation target by adjusting its filter and fusion steps.

---

**Algorithm 1** The Morphological Split-Filter-Fusion Strategy

---

**Input:** Binary coarse mask $\mathcal{M}_{\text{coarse}}$.
**Output:** Processed masks $\mathcal{M}^{\text{msff}}$.
1: $(\mathcal{R}, \text{num\_regions}, \text{stats}) \leftarrow \text{ExtractConnectedComponents}(\mathcal{M}_{\text{coarse}})$ ▷ Get connected regions
2: $A_{\text{total}} \leftarrow \sum_{i=1}^{\text{num\_regions}} \text{stats}[i].\text{area}$ ▷ Total foreground area
3: $\mathcal{S} \leftarrow \varnothing$ ▷ Initialize set of valid regions
4: **for** $i = 1$ to num\_regions **do**
5:      $A_i \leftarrow \text{stats}[i].\text{area}$
6:      **if** $A_i \geq 5$ and $A_i \geq 0.1 \times A_{\text{total}}$ **then**
7:          $\mathcal{S}.\text{append}(i)$ ▷ Keep region if it passes size filter
8:      **end if**
9: **end for**
10: $\mathcal{Q} \leftarrow \mathcal{S}$ ▷ Initialize with size-filtered regions
11: **if** filter by IPQ **then**
12:      **for** $i \in \mathcal{S}$ **do**
13:          $P_i \leftarrow \text{Perimeter}(\mathcal{R} == i)$ ▷ Contour perimeter
14:          $\text{IPQ}_i \leftarrow \frac{4\pi A_i}{P_i^2}$ ▷ Isoperimetric Quotient
15:      **end for**
16:      $\text{IPQ}_{\text{th}} \leftarrow \text{mean}(\text{IPQ}_i \mid i \in \mathcal{S})$
17:      $\mathcal{Q} \leftarrow \{i \in \mathcal{S} \mid \text{IPQ}_i \geq \text{IPQ}_{\text{th}}\}$ ▷ Filter by IPQ
18: **end if**
19: $\mathcal{M}_{\text{merged}} \leftarrow \text{MergeRegions}(\mathcal{Q})$ ▷ Merge based on bounding box and area criteria (Lin et al.)
20: $\mathcal{M}^{\text{msff}} \leftarrow \text{ApplyDomainKnowledge}(\mathcal{M}_{\text{merged}})$ ▷ Use topology, quantity priors
21: **return** $\mathcal{M}^{\text{msff}}$.

---

### 3.3 HIERARCHICAL PROMPT EXCAVATION

Even after extracting anatomically plausible and semantically consistent regions via the MSFF, the resulting coarse segmentations remain imperfect and noisy. Extracting robust and informative prompts from such inputs is still a non-trivial challenge. To address this, we propose a Hierarchical Prompt Excavation strategy that follows a coarse-to-fine pipeline: first localizing the target structure with high confidence, then refining its boundaries using richer spatial context. Unlike prior work that relies on single-type prompts or naive multi-prompt combinations, our approach sequentially leverages point prompts and bounding box prompts, ensuring complementary guidance that mitigates error propagation.

**Point Prompts.** Point prompts aim to extract maximally reliable positional information from noisy coarse segmentations, serving as "anchors" for SAM to distinguish true anatomical foreground from background. The key insight here is that core regions of the coarse foreground exhibit higher confidence, fewer labeling ambiguities, than boundary regions, while background regions in the immediate vicinity of the foreground, rather than distant, irrelevant areas, provide the most informative negative supervision for correcting boundary errors.

We design a distance-aware sampling strategy to select high-confidence positive and negative points from the filtered, ensuring that these prompts align with the true anatomical foreground and background. Positive points should lie in the "core" of the target structure, regions farthest from boundaries, since interior areas in coarse masks generally exhibit higher prediction confidence. We compute the geodesic distance transform within the foreground to identify such interior points:

$$D_{\text{fore}}(x, y) = \min_{(u,v) \in \partial \mathcal{M}_{\text{msff}}} d\left((x, y), (u, v)\right) \tag{1}$$

where $d(\cdot, \cdot)$ denotes the Euclidean distance, and $\partial \mathcal{M}_{\text{msff}}$ is the boundary of the processed coarse mask. The positive point is selected as the pixel with the maximum $D_{\text{fore}}$ value:

$$(x^+, y^+) = \arg \max_{(x,y) \in \mathcal{M}_{\text{msff}}} D_{\text{fore}}(x, y) \tag{2}$$

As for negative points, we aim to identify the most informative and deterministic negative supervisory signal that can help correct boundary errors. Unlike previous works that select the point farthest

from the foreground within the bounding box, which sometimes prevents negative points from selecting completely irrelevant, distant background points. We proposed to select a positive point within the dilated region of the coarse segmentation mask. $\mathcal{M}_{\mathrm{msff}}$ is dilated by $r$ pixels (empirically set to 20) to define a candidate region $\mathcal{D}$, and compute the background distance transform:

$$D_{\mathrm{back}}(x,y) = \min_{(u,v)\in\mathcal{M}_{\mathrm{msff}}} d\left((x,y),(u,v)\right), \quad \text{for } (x,y) \in \mathcal{D} \setminus \mathcal{M}_{\mathrm{msff}} \tag{3}$$

The point with the maximum $D_{\mathrm{back}}$ value within this dilated background is chosen as the negative prompt:

$$(x^-, y^-) = \arg \max_{(x,y)\in\mathcal{D}\setminus\mathcal{M}_{\mathrm{msff}}} D_{\mathrm{back}}(x,y) \tag{4}$$

This strategy ensures that negative points are both far from the foreground and located within anatomically meaningful regions, thereby offering strong and discriminative background signals to correct misclassified boundary areas.

**Box Prompts.** Bounding boxes provide stronger localization signals due to their rich spatial context. After initial localization via point prompts, we employ box prompts to further refine the segmentation. Our box prompt design balances two goals: (1) tightly localizing the target to avoid including excessive irrelevant background, and (2) expanding enough to cover subtle boundary details missing in the coarse mask. For each refined region, we compute the tightest axis-aligned bounding box enclosing the foreground. To ensure sufficient context and avoid truncating relevant structures, we expand this box by a small margin in each direction (left, right, up, and down).

To better leverage medical domain knowledge, we use MedSAM2 (Ma et al., 2025), a medical adaptation of SAM2 (Ravi et al., 2025) fine-tuned on a large corpus of medical images. Since mask prompts were not used during the fine-tuning of MedSAM2, we also abstain from using mask inputs in our enhancement process to maintain consistency and maximize performance.

### 3.4 APPLICATION SCENARIOS

SAMedEnhancer is designed as a versatile, plug-and-play enhancement module that seamlessly integrates into diverse medical image segmentation workflows. Below, we elaborate on its two application scenarios, post-processing and pseudo-label enhancement paradigms, without requiring architectural changes or retraining of the base segmentation model.

**Post-Processing of Model Predictions.** SAMedEnhancer can serve as a model-agnostic post-processing tool to refine coarse segmentation masks generated by any existing model, such as U-Net (Ronneberger et al., 2015), TransUNet (Chen et al., 2024), or other deep learning-based segmenters (Zhou et al., 2018; Cao et al., 2022). The framework takes the initial prediction masks as input, applies the proposed MSFF and hierarchical prompting strategy, and produces enhanced outputs with improved boundary accuracy and anatomical consistency. The method allows existing production pipelines to be enhanced without the need for model retraining or additional annotated data.

**Pseudo-Label enhancement.** Within unsupervised, weakly supervised, or semi-supervised learning frameworks, pseudo-labels are usually generated from unlabeled data, which are then used to train the segmentation model. However, the quality of pseudo-labels directly limits the segmentation performance, as erroneous pseudo-labels propagate during training, leading to biased or unstable models. SAMedEnhancer can be embedded within an iterative training loop to progressively improve pseudo-label quality. Specifically, after the initial pseudo-labels are generated from unlabeled data, SAMedEnhancer enhances these masks using its prompting mechanism. The enhanced pseudo-labels are then used to supervise the model's training. Each iteration further improves pseudo-label quality, creating a "virtuous cycle" where better pseudo-labels train better models, which in turn generate better pseudo-labels.

Table 1: Datasets information.

| Modality | Types | Datasets | Total |
|---|---|---|---|
| Ultrasound | Breast cancer | BUSD | 163 |
| | Thyroid nodule | TNUS | 2626 |
| Colonoscopy | Polyp | Kvasir | 1000 |
| | | CVC-ClinicDB | 612 |
| MRI | Cardiac | ACDC | 100 |

## 4 EXPERIMENT

### 4.1 DATASETS

To comprehensively evaluate the mask enhancement capabilities of SAMedEnhancer, we conduct extensive experiments on several datasets across diverse modalities and settings. Specifically, we employ five publicly available benchmarks: BUSD (Yap et al., 2017), TNUS (Deng et al.), Kvasir (Jha et al., 2019), CVC-ClinicDB (Tajbakhsh et al., 2015), and ACDC (Bernard et al., 2018). A summary of all datasets is provided in Table 1.

We validate the effectiveness of SAMedEnhancer in two key scenarios: as a post-processing step and for pseudo-label enhancement. In the supervised setting, we compare our method against existing post-processing techniques to demonstrate its refinement performance. For semi-supervised learning, we evaluate under varying ratios of labeled to unlabeled data, assessing its ability to boost segmentation performance by both post-processing and improving pseudo-label quality.

### 4.2 EXPERIMENTAL SETUP

Our experimental setup is designed for fair pairwise comparison with each baseline, maintaining identical training conditions. All models, including our baseline U-Net for both fully- and semi-supervised tasks, are trained on a standardized 7:1:2 train/validation/test split. For evaluation, we employ the Dice Similarity Coefficient (DSC) and the 95th percentile Hausdorff Distance (HD95). SAMedEnhancer leverages a pre-trained MedSAM2 model (sam2.1_hiera_tiny) as its foundational segmentation model, implemented in PyTorch. Notably, our method operates on images at their original resolution without a multi-scale strategy.

### 4.3 RESULTS OF FULLY SUPERVISED SEGMENTATION

The post-processing results of fully supervised segmentation on BUSD, TNUS, CVC-ClinicDB, and Kvasir are shown in Table 2. We compare our method with morphology operations (closing and then choosing the largest connected region, denoted as Morphology Op.) (Patil et al., 2017), Level-set (Li et al., 2010), Conditional Random Fields (CRF, Kamnitsas et al. (2017)), and SAMRefiner (Lin et al.). As for SAMRefiner, we first compare using the pretrained weights of ViT-H (denoted as SAMRefiner$^\star$). We also validated the performance of SAMRefiner with medical finetuned weights, MedSAM2 (denoted as SAMRefiner$^\dagger$; the mask prompt is not utilized to align with our implementation of SAMedEnhancer), to compare the performance of SAMRefiner and our SAMedEnhancer under the same pretrained weight setting.

As shown in Table 2, traditional post-processing methods (Morphology Op., Level-set, CRF) exhibit similar performance trends: they marginally improve the HD95 compared to the Baseline, but fail to enhance the DSC. This indicates that these conventional methods can refine boundary smoothness to some extent but lack the capability to optimize segmentation accuracy. For SAM-based methods, SAMRefiner$^\star$ shows modest gains in DSC over the Baseline on TNUS (65.60 vs. 63.75), CVC-ClinicDB (86.46 vs. 85.65), and Kvasir (85.33 vs. 84.71), but its HD95 performance is inferior to traditional post-processing methods. When equipped with medical-specific MedSAM2 weights (SAMRefiner$^\dagger$), the method achieves better performance on CVC-ClinicDB and Kvasir but suffers from significant performance drops on BUSD and TNUS. This suggests that direct adaptation of general SAM refiners to medical data may not guarantee consistent improvements, even with

Table 2: Results of fully supervised segmentation. The best results are **bolded**. $^\star$ denotes that the foundation model is SAM ViT-H (Kirillov et al., 2023; Dosovitskiy et al., 2020). $^\dagger$ denotes that the foundation model is MedSAM2 (Ma et al., 2025).

| Methods | BUSD | | TNUS | | CVC-ClinicDB | | Kvasir | |
|---|---|---|---|---|---|---|---|---|
| | DSC | HD95 | DSC | HD95 | DSC | HD95 | DSC | HD95 |
| Baseline | 79.42 | 43.25 | 63.75 | 76.82 | 85.65 | 23.73 | 84.71 | 58.73 |
| Morphology Op. | 79.29 | 21.78 | 64.09 | 54.62 | 85.44 | 25.60 | 84.36 | 57.41 |
| Level-set | 79.55 | 21.73 | 63.99 | 55.34 | 85.16 | 25.86 | 84.27 | 57.23 |
| CRF | 78.09 | 22.29 | 64.15 | 53.88 | 85.06 | 25.92 | 83.95 | 57.89 |
| SAMRefiner$^\star$ | 78.79 | 33.73 | 65.60 | 68.84 | 86.46 | 22.08 | 85.33 | 56.54 |
| SAMRefiner$^\dagger$ | 69.95 | 45.74 | 61.32 | 74.72 | 87.17 | **20.52** | 85.78 | 55.46 |
| SAMedEnhancer | **84.75** | **16.57** | **67.87** | **49.98** | **87.28** | 20.78 | **86.41** | **52.96** |

Table 3: Semi-supervised segmentation results (DSC). **Post**: SAMedEnhancer for post-processing; **PE**: for pseudo-label enhancement.

| Methods | 3/70 labeled | | | | 7/70 labeled | | | |
|---|---|---|---|---|---|---|---|---|
| | LV | MYO | RV | Ave. | LV | MYO | RV | Ave. |
| CPS | 42.11 | 64.12 | 78.38 | 61.54 | 71.47 | 82.25 | 89.68 | 81.13 |
| CPS+Post | 58.34 | 75.24 | 82.44 | 72.01 | 76.51 | 85.26 | 88.04 | 83.27 |
| CPS+PE | 52.53 | 71.73 | 82.04 | 68.77 | 73.76 | 83.89 | 88.23 | 81.96 |
| CPS+PE+Post | 58.72 | 80.01 | 81.55 | 73.43 | 75.99 | 84.73 | 87.10 | 82.61 |

domain-finetuned weights. In contrast, our SAMedEnhancer consistently outperforms all competing methods across both metrics and all datasets. It achieves the highest DSC values on every dataset, representing notable gains of 5.33, 4.12, 1.63, and 1.70 percentage points over the Baseline, respectively. These results demonstrate that our SAMedEnhancer effectively integrates the strengths of medical image understanding and foundation model refinement, achieving both high segmentation accuracy and precise boundary localization.

Visualizations in Figure 2 (shown in Appendix) further corroborate these quantitative gains: unlike traditional methods that over-smooth fine anatomical details or leave residual artifacts, and SAMRefiner which often retains boundary from coarse inputs, SAMedEnhancer precisely recovers ambiguous edges and eliminates spurious fragments while preserving structural integrity.

### 4.4 RESULTS OF SEMI-SUPERVISED SEGMENTATION

We validate the efficacy of SAMedEnhancer in semi-supervised segmentation on the ACDC dataset, a benchmark for cardiac MRI segmentation targeting three key anatomical structures: Left Ventricle (LV), Myocardium (MYO), and Right Ventricle (RV). We adopt the Cross Pseudo Supervision (CPS) framework as our baselin (Chen et al., 2021), and systematically evaluate the performance of SAMedEnhancer in two integration modes, *Post-processing (Post)* and *Pseudo-label Enhancement (PE)*, as well as their combined use (*PE+Post*). Experiments are conducted under two low-label regimes: 3 labeled samples (3/70) and 7 labeled samples (7/70), with results reported in Table 3. As evident from Table 3, integrating SAMedEnhancer consistently yields substantial performance gains across both label ratios, demonstrating its ability to alleviate annotation dependency in semi-supervised learning.

### 4.5 ABLATION STUDY AND DISCUSSION

To dissect the contributions of each core component in SAMedEnhancer, including the Morphological Split-Filter-Fuse (MSFF) module and the hierarchical Point/Box prompts, we conduct systematic ablation experiments on the BUSD and Kvasir datasets. We evaluate variants of our framework by ablating individual or combined components, as summarized in Table 4.

Table 4: Ablation Study.

| MSFF | Point | Box | Mask | BUSD | | Kvasir | |
|---|---|---|---|---|---|---|---|
| | | | | DSC | HD95 | DSC | HD95 |
| | | | | 79.42 | 43.25 | 84.71 | 58.73 |
| | ✓ | | | 83.81 | 16.12 | 85.65 | 52.49 |
| | | ✓ | | 68.38 | 48.28 | 85.44 | 58.05 |
| | | | ✓ | 9.33 | 125.12 | 51.56 | 159.47 |
| | ✓ | ✓ | | 84.59 | 16.84 | 86.24 | 52.62 |
| ✓ | | | | 81.51 | 16.58 | 84.46 | 57.2 |
| ✓ | ✓ | | | 84.05 | 16.29 | 85.71 | 53.85 |
| ✓ | ✓ | ✓ | | 84.75 | 16.57 | 86.41 | 52.96 |

**Baseline and Prompts Effectiveness.**  Among single-prompt variants, point prompts alone deliver substantial gains over the baseline, boosting DSC by 4.39 on BUSD and 0.94 on Kvasir, while reducing HD95 by 27.13 and 6.24, respectively. This confirms that our distance-aware point sampling, selecting high-confidence interior positives and informative nearby negatives, which effectively anchors SAM to true anatomical foreground/background, even without additional components. Box prompts alone perform poorly, with DSC dropping to 68.38 (BUSD) and 85.44 (Kvasir). Tight bounding boxes derived from unfiltered coarse masks tend to inherit boundary errors, while expansion margins fail to compensate for noisy initial regions without point-based anchor guidance. Mask prompts alone are catastrophic, consistent with our design choice: MedSAM2 was not fine-tuned on mask prompts, so feeding noisy coarse masks directly amplifies errors rather than guiding refinement. Combining Point and Box prompts (without MSFF) further improves performance. This synergy arises because point prompts provide precise positional anchors, while box prompts supply spatial context to capture subtle boundary details, addressing the limitations of either prompt type in isolation.

**Impact of the MSFF Module.**  The MSFF module, which cleans coarse masks into anatomically plausible regions, contributes meaningfully when paired with prompts. MSFF alone offers modest DSC gains (81.51 vs. 79.42 on BUSD) but drastically reduces HD95 (16.58 vs. 43.25), confirming its role in filtering noise and improving boundary consistency. Combining MSFF and Point prompts outperforms both MSFF alone and Point prompts alone. By eliminating spurious fragments and merging coherent regions, MSFF ensures point prompts are sampled from semantically reliable areas, reducing noise-induced bias.

**Full Framework Efficacy.**  The complete SAMedEnhancer (MSFF + Point + Box) achieves the best performance across all metrics. This confirms that all components work synergistically: MSFF provides a clean foundation for prompting, Point prompts anchor SAM to high-confidence regions, and Box prompts supply contextual guidance, collectively resisting error propagation and refining boundaries.

## 5 CONCLUSION

This work introduces SAMedEnhancer, a generic, model-agnostic framework that adapts the medical-tailored MedSAM2 for robust segmentation enhancement. At its core, our morphology-aware prompting strategy—comprising the Morphological Split-Filter-Fuse (MSFF) module and Hierarchical Prompt Excavation—effectively filters noisy coarse masks and generates reliable point/box prompts, guiding SAM to correct boundary errors without propagating inaccuracies. Extensive experiments across five datasets spanning ultrasound, colonoscopy, and MRI modalities demonstrate that SAMedEnhancer consistently outperforms traditional post-processing techniques and SAM-based baselines in both fully supervised (as a post-processor) and semi-supervised (for both post-processing and pseudo-label enhancement) settings, achieving superior DSC and HD95. As a plug-and-play tool requiring no task-specific retraining, SAMedEnhancer offers a versatile solution to boost segmentation quality across diverse medical imaging tasks, reducing annotation dependency and advancing the practical deployment of automated segmentation systems.

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

# A APPENDIX

## A.1 ETHICS STATEMENT

This study complies with ethical principles in medical image research. No human subjects or animal experimentation were conducted. All medical image datasets (BUSD, TNUS, Kvasir, CVC-ClinicDB, ACDC) were sourced from public repositories with authorized research access, adhering to data privacy policies. We ensured no personally identifiable information was used and took steps to avoid biased sampling or discriminatory outcomes during data processing and analysis. The research process maintained transparency in handling data and conducting experiments without privacy or security risks.

## A.2 REPRODUCIBILITY STATEMENT

To ensure the reproducibility of our research findings and facilitate further exploration in the field of medical image segmentation enhancement, we solemnly commit to publicly releasing all relevant datasets and code upon the formal acceptance of this manuscript.

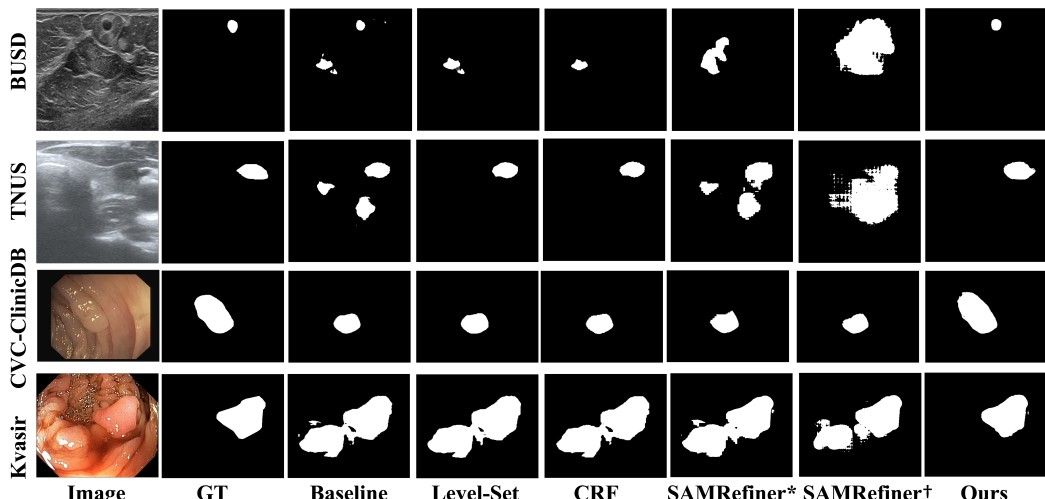

Figure 2: Visualization of different methods.

## A.3 LLM USAGE

Large Language Models (LLMs) assisted in manuscript preparation, focusing on language polishing: refining sentence structure for clarity, enhancing readability of technical explanations, and conducting grammar verification.

Notably, LLMs were not involved in formulating research ideas, designing methodologies, or analyzing experimental data. All scientific concepts, model innovations, and result interpretations were developed by the authors. The LLM's role was strictly limited to improving linguistic expression, without influencing scientific content.

The authors assume full responsibility for the manuscript's content, including LLM-aided text. We verified that LLM-generated or polished text meets ethical standards, avoids plagiarism, and accurately represents the research findings.

