# OpenReview forum: "Generic Medical Image Segmentation Enhancement by Adapting Segment Anything Model"
_ICLR.cc/2026/Conference — ICLR 2026 Conference Withdrawn Submission_

### Official Review · Reviewer_TTGJ · 2025-10-17

**Soundness:** 2
**Presentation:** 2
**Contribution:** 2
**Rating:** 2
**Confidence:** 5

**Summary:**

This paper proposes SAMedEnhancer, a model-agnostic framework for enhancing medical image segmentation masks by leveraging the Segment Anything Model. The method operates by converting coarse segmentation masks from any model into robust morphological prompts, consisting of filtered connected components and hierarchical positive/negative point and bounding box prompts.

**Strengths:**

The approach is model-agnostic and can be applied as a post-processing enhancement step to any existing segmentation pipeline without retraining. The paper claims intent to release code/datasets upon acceptance.

**Weaknesses:**

My major concern is the motivation of the paper. When sufficient training data is available, deep learning models outperform SAM variants, though they require relatively large datasets for training. SAM variants typically demonstrate suboptimal performance across most medical image analysis tasks, and the extent to which their performance can be significantly improved remains questionable. Additionally, there is a lack of comparisons between SAM variants and SOTA task-specific models such as nnU-Net.

Other Concerns
Recent domain-generalizable or robust SAM adaptation methods is not included as baselines in experimental results, weakening the empirical claim of proposed method.
The method s generalization to highly multi-class or multi-label settings, or to complex 3D (volumetric) data, is not discussed or tested. All experiments focus on 2D slice data, leaving the claimed plug-and-play generalization only partially substantiated by evidence. As most “medical image segmentation” tasks are multi-target segmentation adopted on 3D images like CT and MRI.
The results in Tables without any reporting of variance or statistical significance. No confidence intervals, standard deviations, or formal significance tests are presented, due to possible high variance in medical segmentation tasks especially with semi-supervision and small data splits.
Some steps of the mathematical formulation are underspecified, including how the filter and fusion rules are adapted per task what concrete domain knowledge or priors are needed for reliable MSFF operation? Some tables lack context, and table formatting across the paper is occasionally inconsistent. Figure 2 is referenced only in the appendix but should arguably be part of the main results for less disruption to the narrative.

**Questions:**

Refer to Weakness

---

### Official Review · Reviewer_c4Td · 2025-10-25

**Soundness:** 3
**Presentation:** 2
**Contribution:** 2
**Rating:** 2
**Confidence:** 5

**Summary:**

This work uses Segment Anything Model 2 (SAM2) and its medical counterpart MedSAM2 to enhance segmentation masks for medical use-cases. Towards this end, they propose a stage-wise refinement process. This includes a filter for removing noisy regions and combining disconnected regions called MSFF, followed by Prompt Excavation that extracts point and box-based prompts. These are input to the pretrained MedSAM2 model for refined masks. Experiments show improved performance over UNet on multiple datasets

**Strengths:**

1) The paper presents an application of SAM-like foundation models as refinement modules and discusses valid potential applications, like improving semi-supervised learning and post-processing.

2) Prompt selection in the Excavation phase is done using geodesic distance to isolate useful positive and negative point prompts. This allows SAM to pinpoint on the area of interest

3) MSFF module allows the insertion of domain knowledge about the object of interest that can be used to filter out noise.

**Weaknesses:**

1) Limited Experimentation: In line 156, the authors claim that SAMedEnhancer is model agnostic. Yet, the authors only show results on UNet as a baseline. More baselines have to be included to uphold this claim. In addition, many methods adapt SAM directly for the segmentation task and get significantly better results than UNet. Even without SAM, many methods like nnUNet, TransUNet and UNext exist that show significantly better performance than UNet. Thus, I believe the choice of only UNet as a baseline is insufficient to gauge the effectiveness of the method.

2) The paper does not discuss the potential limiting cases of the method. One important possibility that I can think of is if the baseline model generates a mask that does not cover the object of interest at all. In such a case, all the following modules will fail to extract a correct prompt.

3) The method has lots of hyperparameters that necessitate the need for engineering them correctly for various applications. For example, the size filter which is set to 5 (why?), the IPQ threshold, margin for the box prompt and so on.

4) The paper requires a lot of minor formatting changes as follows:
(a) The related work section should contain at least 3-4 lines about the SAMRefiner method, which seems like a direct predecessor to the method and an important comparison.
(b) Tables 3 and 4 should indicate the best values in bold.
(c) The methods in Table 2 should be cited in either the table rows or the caption itself.
(d) Cite the existing works for the single-type prompts and multi-prompt combinations in line 248
(e) Missing third stage description in 3.1

**Questions:**

1) What is the effect of IPQ in the overall performance of the MSFF? The ablations treat MSFF as a whole component, but I am interested to know the individual effect of domain knowledge filters like IPQ over general area-based morpological operations.

2) In line 20 of the algorithm, the authors mention that domain knowledge can be applied. What are some examples other than IPQ?

3) Please refer to the weaknesses. My main concern is (1)Limited Experimentation

---

### Official Review · Reviewer_cqBC · 2025-10-29

**Soundness:** 2
**Presentation:** 3
**Contribution:** 2
**Rating:** 2
**Confidence:** 5

**Summary:**

1. Propose SAMedEnhancer, the first framework to effectively leverage SAM for generic medical image segmentation enhancement by morphological analysis and hierarchical prompt excavation. It operates in a plug-and-play manner, refining coarse outputs from any segmentation model without retraining.
2. Establish a large-scale evaluation benchmark for medical image segmentation enhancement, covering multiple datasets, modalities, and supervision settings, to facilitate future research in this under-explored area.
3. Introduce a novel semi-supervised learning strategy that integrates iterative pseudolabel refinement with SAMedEnhancer, demonstrating significant performance gains and reduced reliance on annotated data.

**Strengths:**

1. The methods are well-described. Authors provided pseudo codes and equations to help authors to understand their methods.
2. The related works are well-described.

**Weaknesses:**

1. The overall contributions is low. Authors proposed a SAMedEnhancer. However, the core component of this method is SAM which was proposed by others, and many SAM-related works have been proposed and explored. The improvement over SAM is incremental, and authors did not make fundamental contributions to this field.
2. Th evaluation is not sufficient. Authors evaluated their methods on five datasets. However, they did not evaluate their methods in CT datasets. CT is the most widely used imaging modality in clinical practice. Additionally, they evaluated their methods on single organ segmentation, but authors did not evaluate their methods on multi-organ segmentation. Currently multi-organ segmentation is more challenging, and implementing multi-organ segmentation will make more scientific contributions.
3. The comparison with other methods is not sufficient. Authors only compared their methods with four other methods which were proposed several years ago.
4. The qualitative evaluation is not sufficient. Authors compared the segmentation results of their methods and other methods, but they did not provide qualitative evaluation about how their methods work.

**Questions:**

1. The overall contributions is low. Authors proposed a SAMedEnhancer. However, the core component of this method is SAM which was proposed by others, and many SAM-related works have been proposed and explored. The improvement over SAM is incremental, and authors did not make fundamental contributions to this field.
2. Th evaluation is not sufficient. Authors evaluated their methods on five datasets. However, they did not evaluate their methods in CT datasets. CT is the most widely used imaging modality in clinical practice. Additionally, they evaluated their methods on single organ segmentation, but authors did not evaluate their methods on multi-organ segmentation. Currently multi-organ segmentation is more challenging, and implementing multi-organ segmentation will make more scientific contributions.
3. The comparison with other methods is not sufficient. Authors only compared their methods with four other methods which were proposed several years ago.
4. The qualitative evaluation is not sufficient. Authors compared the segmentation results of their methods and other methods, but they did not provide qualitative evaluation about how their methods work.

---

### Official Review · Reviewer_5xih · 2025-11-01

**Soundness:** 1
**Presentation:** 3
**Contribution:** 1
**Rating:** 2
**Confidence:** 4

**Summary:**

This paper presents SAMedEnhancer, a model-agnostic framework that leverages the Segment Anything Model (SAM) to enhance coarse medical image segmentation masks without retraining. It introduces a Morphological Split-Filter-Fuse (MSFF) module to clean noisy masks and a Hierarchical Prompt Excavation strategy to generate robust point and box prompts that guide SAM for precise boundary refinement. Experiments across multiple modalities show consistent improvements in Dice and Hausdorff Distance, demonstrating that SAMedEnhancer effectively boosts segmentation accuracy and serves as a plug-and-play enhancement tool for both fully and semi-supervised settings.

**Strengths:**

- The method is completely model-agnostic and plug-and-play, requiring no retraining of either the base segmentation model or SAM.
- The combination of the MSFF module and hierarchical point/box prompts enables effective noise reduction and preserves structural consistency.
- The approach is thoroughly tested across multiple imaging modalities and tasks, including semi-supervised pseudo-label enhancement.
- It consistently outperforms traditional post-processing methods such as CRF, level-set, and morphology operations in both Dice and HD95 metrics.

**Weaknesses:**

- The framework mainly combines morphological analysis with improved SAM prompting strategies, which is practical but conceptually incremental. The methodological novelty is somewhat limited, as it extends existing SAM-based refinement pipelines rather than introducing new theoretical insights or learning principles.
- The ablation study shows only modest gains for individual modules, and the overall improvement may partly stem from the introduction of additional input information (e.g., extra point, box, or mask prompts), which inherently provides more spatial cues rather than demonstrating a fundamentally stronger reasoning mechanism.
- The experimental datasets are relatively small and may not adequately support claims of generalization across diverse medical modalities or large-scale clinical settings.
- From the visual examples in Figure 2, the improvements appear marginal; SAM itself may already achieve comparable results in a zero-shot manner. The paper does not provide a clear discussion or quantitative comparison to isolate such effects.
- The source of the *coarse masks* used as inputs is unclear. It is important to specify whether these masks are generated by real segmentation models or synthetically constructed, as this directly affects the validity and applicability of the proposed enhancement scenario.

**Questions:**

I have the following questions for author(s):
- How are the coarse masks obtained—are they real model outputs or synthetic approximations?
- How does SAMedEnhancer outperform zero-shot SAM with similar prompts, and is there a quantitative comparison?
- The ablation gains seem modest—can the authors justify each component’s necessity or report significance analyses?
- How do the authors separate the benefit of extra prompt information from the algorithmic contribution?
- Can results on larger or more diverse datasets (e.g., multi-organ CT or 3D MRI) be provided to support generalizability?
- What is the computational cost of SAMedEnhancer, and how practical is it for large-scale or real-time use?

---

### Note · Authors · 2025-11-13

I have read and agree with the venue's withdrawal policy on behalf of myself and my co-authors.